# The Adult and Larva of a New Species of the Genus *Dila* (Coleoptera, Blaptinae, Blaptini) from Himalayas, with Molecular Phylogenetic Inferences of Related Genera of the Blaptini [note 1]

**DOI:** 10.3390/insects14030284

**Published:** 2023-03-13

**Authors:** Xiu-Min Li, Baoyue Ji, Juan Tian, Guo-Dong Ren

**Affiliations:** Key Laboratory of Zoological Systematics and Application of Hebei Province, College of Life Sciences, Institute of Life Science and Green Development, Hebei University, Baoding 071002, China; baoyueji2000@163.com (B.J.); juantian521@163.com (J.T.)

**Keywords:** *Dila*, new species, morphological description, DNA sequence

## Abstract

**Simple Summary:**

The genus *Dila* Fischer von Waldheim of 1844 belongs to the tribe Blaptini Leach of 1815 within the subfamily Blaptinae. Comprising 22 species, this genus is broadly distributed in Southeastern Turkey, Iraq, Iran, and Central Asia. However, *Dila* is poorly known in China; only three species have been described from Jomda of Xizang, Dêgê, Sichuan and Xinjiang, China. In this study, the larva was associated with the adults using a combined molecular dataset. The larva and adult males and females of the new species are described and illustrated from the southwestern Himalayas. To date, the phylogenetic relationships within Blaptini is not well supported by molecular evidence. A preliminary phylogeney was hypothesized and discussed based on a molecular dataset with seven related genera of the tribe Blaptini. The monophyly of the subtribe Dilina and the taxonomic status of *D. bomina* are briefly discussed.

**Abstract:**

In this study, a new species of the genus *Dila* Fischer von Waldheim, 1844, *D. ngaria* Li and Ren sp. n., was described from the southwestern Himalayas. The adult and larva were associated using molecular phylogenetic analyses based on fragments of three mitochondrial and one nuclear gene fragment (COI, Cytb, 16S and 28S-D2). Additionally, a preliminary phylogenetic tree was reconstructed and discussed based on a molecular dataset with seven related genera and 24 species of the tribe Blaptini. Meanwhile, the monophyly of the subtribe Dilina and the taxonomic status of *D. bomina* Ren and Li, 2001 are discussed. This work provides new molecular data for phylogenetic studies on the tribe Blaptini in the future.

## 1. Introduction

The genus *Dila* Fischer von Waldheim of 1844 belongs to the tribe Blaptini Leach of 1815 within the subfamily Blaptinae. The type species *Dila laevicollis* Gebler, found in 1841, is broadly distributed in Afghanistan, Kyrgyzstan, Kazakhstan, Tajikistan, Turkmenistan, Turkmenistan and Xinjiang, China [1]. Chigray (2019) revised the genus, considered *Caenoblaps* as a junior synonym of *Dila*, transferred four species from *Caenoblaps* to *Dila*, and described three new species from Turkey and Iran [2]. To date, the genus *Dila* comprises 22 species distributed in the mountain regions of Southeastern Turkey, Iraq, Iran, and Central Asia (Tien-Shan, Pamir-Alay, Western Himalaya, and Tibet) [2,3,4]. These species were described by Baudi di Selve (one species), Blair (one species), Chigray (three species), Gebler (one species), Kaszab (three species), König (one species), Reitter (five species), Ren (two species), Schuster (four species), and Semenov and Bogatshev (one species) in 1841–2019 [5,6,7,8,9,10,11]. In addition, Chigray proposed that the Iranian species *Blaps platythorax* Gemminger, found in 1870 (described from Shiraz), probably belongs to the genus *Dila* [2]. Although *Dila* exhibits species richness, only three species are currently known from China (*D. laevicollis* Gebler, 1841, *D. bomina* Ren and Li, 2001, and *D. rugelytra* Ren, 2016). Skopin noted the close similarity of the larvae of *Dila* to the primitive larvae of *Blaps* based on similar habitats [12]. However, the morphological characters of larva of *Dila* have not been described yet.

In this study, the larva and adult male and female of a new species are described based on morphological and molecular evidence. In addition, we constructed a molecular phylogeny for seven related genera of the tribe Blaptini and four outgroup taxa, to verify the taxonomic status the new species. We also provided molecular data for the phylogenetic studies of the tribe Blaptini in the future.

## 2. Materials and Methods

### 2.1. Morphological Examination

In total, 99 specimens from three sampling localities were examined for this study and have been deposited at the Museum of Hebei University, Baoding, China (MHBU). The figures were taken with three distinct imaging systems: (a) a Canon EOS 5D Mark III (Canon Inc., Tokyo, Japan) connected to a Laowa FF 100 mm F2.8 CA-Dreamer Macro 2× or Laowa FF 25 mm F2.8 Ultra Macro 2.5–5× (Anhui Changgeng Optics Technology Co., Ltd., Hefei, China); (b) a Leica M205A stereomicroscope equipped with a Leica DFC450 camera (Leica Microsystems, Singapore, Singapore), which was controlled using the Leica application suite 4.3; (c) a JVC^®^ KY–F75U (JVC Kenwood, Long Beach, CA, USA) digital camera attached to a Leica Z16 APO dissecting microscope (Leica Microsystems, Buffalo Grove, IL, USA) with an apochromatic zoom objective and motor focus drive, using a Syncroscopy^®^ Auto-Montage System (Synoptics, Cambridge, UK) and software. Multiple images were used to construct the final figures. Images were illuminated with either an LED ring light attached to the end of the microscope column, with incidental light filtered to reduce glare, or by a gooseneck illuminator with bifurcating fiberoptics; image stacks were white-balance-corrected using the software of the imaging system (Synoptics, Cambridge, UK). Label data are presented verbatim. A double slash (//) separates text on different lines of label, authors’ remarks are enclosed in brackets “[]”.

Type material: *Dila laevicollis*: lectotype: ♂, labelled: “Camp. Kirgis.”, “Lectotypus, *Dila laevicollis* Gebl”; paralectotype: 1 ♀: “Camp. Kirgis.”, “*laevicollis* Gebl, Camp. Kirg.”, “Hololectotypus”.

### 2.2. Taxon Sampling, DNA Extraction, PCR Amplification, and Sequencing

DNA was extracted from leg muscle tissue of adults using Insect DNA isolation Kit (BIOMIGA, Hangzhou, China) following the manufacturer’s protocols. Fragments of three mitochondrial molecular markers (cytochrome coxidase subunit I, COI; cytochrome b, Cytb; 16S ribosomal RNA, 16S), and of one nuclear molecular marker (28S ribosomal RNA domain D2, 28S–D2) were amplified and sequenced. The primers and annealing temperature are shown in Table 1. 

The profile of the PCR amplification consisted of an initial denaturation step at 94 °C for 4 min, 35 cycles of denaturation at 94 °C for 1 min, annealing for 1 min, and extension at 72 °C for 1 min, and a final 8 min extension step at 72 °C. PCR was performed using TaKaRa ExTaq (TaKaRa, Dalian, China). PCR products were subsequently checked by 1% agarose gel electrophoresis, and sequencing was performed at GENERAL BIOL Co., Ltd. (Chuzhou, China).

### 2.3. Phylogenetic Analyses

Phylogenetic analyses were carried out on the concatenated dataset under maximum likelihood (ML). For ML analyses, we used IQ-TREE v1.6.6 [17] as implemented in the dedicated IQ-TREE web server (http://iqtree.cibiv.univie.ac.at/, accessed on 1 June 2022). The ML tree was inferred under an edge-linked partition model for 5000 ultrafast bootstraps (1000 replicates) [18]. The phylogenetic tree was visualized in Figtree v.1.4.4. (http://tree.bio.ed.ac.uk/software/figtree, accessed on 1 June 2022). Altogether, 240 sequences from 62 individuals of 24 species of 7 genera (*Ablaps*: 1 species, *Blaps*: 6 species, *Coelocnemodes*: 2 species, *Dila*: 3 species, *Gnaptor*: 1 species, *Nalepa*: 7 species, and *Prosodes*: 4 species) of the tribe Blaptini were used in this phylogenetic analysis. To complete this sampling, we also used previously published sequences from one specimen of *Blaps* (*B. rhynchoptera*) and one specimen of *Gnaptor* (*G. spinimanus*). Meanwhile, we used sequences from 4 individuals of the tribe *Platyscelidini* as outgroup, which has been recovered as a close relative of Blaptini. Detailed information for all the samples used in this study are provided in Appendix A.

## 3. Results

### 3.1. Morphological Study and Diagnosis

#### 3.1.1. Key to Species of the Genus Dila Fischer von Waldheim, 1844, from China (Based on Males)

1.The body is slender (Figure 2A) and elongated; the parameres are finger-shaped… 2

-The body is robust and weakly elongated; the parameres are cone-shaped … …   … 3

2.The pronotum  is wider than it is  long and almost cordiform, with  one tooth on theventral margin of the profemur …  … … … …   …   …   …   *D. laevicollis* Gebler, 1841

-The pronotum length and width are nearly equal, and the lateral margins show slightarcuate  narrowing  to  the  anterior  margin,  with  obtuse-angled  prominence of  theprofemur  …  … … … …  …  …   …   …   …  …   …  …   … *D. ngaria* Li and Ren, sp. n.

3.The  pronotum is wider  than it is  long and almost cordiform; with  two teeth on theventral  margin of  the profemur,  without  one  tooth on  the ventral  margin  of  themesofemur  …  …   …  … … …  …   …  …   …    …  …  … *D. bomina* Ren and Li, 2001

-The  pronotum  length   and   width   are  nearly   equal,   the    lateral  margins  showslight  arcuate  narrowing  to   the  anterior  margin,  with   one  tooth  on  the  ventralmargin   of   the  profemur   and   without   a   tooth  on  the  ventral   margin   of   themesofemur   …    …   …    …  … …  …    …  …   …    …   …     … *D. rugelytra* Ren, 2016

***Dila ngaria* Li and Ren sp. n**.

**Type locality**. China, Xizang, Tsada County, Xiangzi. 

Type specimens (Adults). Holotype: ♂, with the following labels: “2015.VIII.24//Xiangzi Township, Tsada County, Xizang, China//Guo-Dong Ren et al.//Museum of Hebei University”//“31°32.038′N//79°59.020′E//Elev.4420 m//Museum of Hebei University”. Paratypes: 15 ♂ 7 ♀ [1 ♂ 1 ♀ in ethanol] (MHBU), same data as holotype; 2 ♀ in ethanol (MHBU), labeled “2018.VIII.11//Tuolin Township, Tsada County, Xizang, China//Xing-Long Bai et al.//Museum of Hebei University]”//“31°32.038′N//79°59.020′E//Elev. 4496 m//Museum of Hebei University”; 22♂30♀[9♂11♀ in ethanol] (MHBU), labeled “2022.VII.11//Burang Township, Burang County, Xizang, China//Guo-Dong Ren et al.//Museum of Hebei University]”//“30°16.5852′N//81°11.4735′E//Elev. 4006 m//Museum of Hebei University”.

**Description** (Figure 2C). Body length 20.0–22.0 mm, width 6.5–7.5 mm; black, nearly cylinder-shaped, elongated. 

**Head** (Figure 1A,B). Anterior margin of epistome emarginated. Lateral margins of epistome straight; lateral margins of head with indistinct emargination between epistome and genae; head widest at eye level; mentum transverse, with elliptical lateral sides; coarsely punctate and slightly impressed in middle of anterior edge; antennae long, filiform, reaching beyond pronotal base when directed backwards, antennomeres Ⅲ–Ⅶ long and cylindrical, Ⅷ–Ⅹ oval, Ⅺ spindle (Figure 1C). Ratio of length/width of antennomeres Ⅱ–Ⅺ 4.0 (8.0): 40.0 (10.0): 22.0 (10.0): 24.0 (9.0): 23.0 (9.0): 26.0 (9.0): 12.0 (11.0): 13.0 (11.0): 13.0 (11.0): 16.0 (10.0).

**Thorax** (Figure 1D). Long and wide nearly equal, 1.6 times as wide as head; widest at middle, lateral margins more arcuately narrowing to anterior margin than to base; posterior margin straight; anterior angles obtuse, posterior ones weakly obtuse or rectangular; ratio of width at anterior margin to base 30: 42; disc strongly convex, smooth, surface with dense punctation. 

**Abdomen**. Elytra elongate, 2.1 times as long as wide, widest at middle, weakly arcuated, 3.1 times as long and 1.35 times as wide as pronotum, 2.2 times as wide as head; strongly convex on disc, elytral surface smooth, with dense punctures, apex of elytra with attenuate sloping, obtuse; abdomen ventrites 1st–3rd with transverse/longitudinal wrinkles, abdominal ventrites 4th–5th with dense punctures and simple particles.

**Legs** (Figure 1E–J). Legs slender, profemur extension at middle, protibiae nearly straight, inner mesotibiae weakly curved, upper apical spur spinal; ventral surface of protarsomeres Ⅰ–Ⅳ, meso-, metatarsomeres Ⅰ–Ⅲ with hairy brush. Ratio of length of pro–, meso– and metatibiae 33: 31: 41, that of metatarsomeres I–Ⅳ 8.0 (4.0): 5.0 (3.0): 5.0 (3.0): 11.0 (3.0).

**Male genitalia** (Figure 1K–M). Aedeagus finger-shaped, length 4.5–4.6 mm, width 0.7–0.8 mm; parameres length 1.7–1.8 mm and width 0.5–0.6 mm, apex obtuse; parameres widest at base, evenly narrowed to acute apex in dorsal view; parameres almost in a straight line up to apex in lateral view; rods of spiculum gastrale merged at apex, forming long common stem (Figure 1L).

**Sexual dimorphism** (Figure 2D). Body nearly cylinder-shaped, elongated. Length 21.0–23.0 mm, width 7.5–8.0 mm; body wider than male; head 1.4 times as wide as interocular distance, pronotum 1.1 times as wide as long, elytra twice as long as wide. Antennae shorter than male, antennomeres Ⅲ–Ⅶ long and cylindrical, Ⅷ–Ⅹ oval, Ⅺ spindle; pronotum and elytra convex; protibiae nearly straight, inner mestibiae weakly curved, upper apical spur spinal. 

**Etymology**. This species is named from the type locality (located in Ngari). 

**Distribution**. China, Xizang.

**Diagnosis**. The new species is morphologically very similar to *Dila laevicollis*, but can be distinguished from the latter by the following male character states: (1) the body length is 20.0–22.0 mm, elongated, nearly cylinder-shaped, (the body length is 29.0–32.0 mm, elongated, cylinder-shaped in *D. laevicollis*); (2) the pronotum length and width are nearly equal, and the lateral margins show slight arcuate narrowing to the anterior margin (the pronotum is wider than it is long and is almost cordiform in *D. laevicollis*); (3) there is one tooth-like process on the ventral margin of the profemur (with obtuse-angled prominence of the profemur in *D. laevicollis*).

### 3.1.2. Larva

**Body** (Figure 3A–C). Mature larvae length 20.0–22.0 mm, width 2.5–3.0 mm. Body orthosomatic with sides subparallel, subcylindrical with sharp tail-end; moderately sclerotized; yellowish brown; body vestiture consisting of short to moderately elongate; vestiture smooth; median line is obvious at the first 3 segments, Y–shaped; covered with pairs long setae on each tergite (besides the last segment); posterior border of each segment with dark brown longitudinal stripes. 

**Head** (Figure 3D,E). Prognathous, slightly narrower than width of prothorax, slightly convex dorsally, sides rounded (Figure 3D); labrum transverse, weakly emarginate, apical margin with sparse of setae; mandibles well developed, left and right sides symmetrical, with 2 pairs short setae, elongate anterior extension; clypeus transverse, trapezoid-shaped; lateral margins arcuately narrowing to middle, then arcuately narrowing to anterior margin; disc feebly convex, anterior margin slightly linear, and with 4 long erect clypeal setae; epicranial stem Y-shaped (Figure 3E); frons and epicranial plate slightly convex, smooth, with sparse and fine punctures, lateral margins with two row dense long hairs, the center of anterior rim of frons with 4 pairs hairs (Figure 3E); maxillary palps 3-segmented, subcylindrical, with cone-shaped at peak; Ⅰ widest, Ⅱ and Ⅲ with same length, twice as long as Ⅰ (Figure 3E); labial palps 2-segmented, short, Ⅱ cone-shaped (Figure 3E); mentum convex, U-shaped, base of mentum straight, prementum with 2 long hairs; submentum trapezoid-shaped, anterior margin nearly straight, posterior margin curved; antennae well developed, 3-segmented with dome-like at peak, approximate cylindrical, shorter than length of head, segment Ⅰ noticeably wider; Ⅲ is longest, about 1.5 times as long as the Ⅱ, and 0.9 times as wide as Ⅱ. 

**Thorax** (Figure 3D,E). Thoracic segmentation C-shaped in dorsal view, sides parallel, widest in middle, with transverses plicae; each thoracic tergum with long slender setae on sides of anterior and posterior margins; pronotum longest, about 0.8 times as long as sum of meso- and metanotum; mesonotum shortest.

**Legs** (Figure 3E). Legs well developed; prothoracic leg noticeably stronger, much thicker than meso- and metathoracic ones; profemur, femora and tibia with sparse long setae; pro- and mesotarsungulus strongly sclerotized sharp claw-like, profemora and protibia gradually narrowing towards apex, profemora about 0.8 times length of protibia; metatarsungulus highly ossified hook-like.

**Abdomen** (Figure 3A–C,F,G). Approximately 3.6 times as long as thorax; segments Ⅰ–Ⅷ subcylindrical with transverses plicae, faintly rugose, with sparse elongate setae ventrally; tergum nearly same long and segments Ⅶ–Ⅷ narrower than others, pygopods subconical in dorsal view, marginal with two rows of Ⅶ–Ⅸ short spines, urogomphi with 2 short spines at about middle; urogomphi suddenly upturned to apex in lateral view; surface of convex disc with sparse long setae in ventral view (Figure 3F,G). 

**Spiracles** (Figure 3B). A well-developed pair, round thoracic spiracles, situated ventrolaterally on anterolateral margins of terga Ⅰ–Ⅷ.

### 3.2. Phylogenetic Relationships

The final, concatenated dataset was 2289 bp long, including sequences of seven genera from Blaptini (COI, 744 bp; Cytb, 579 bp; 16S, 503 bp; 28S-D2, 463 bp). IQ-TREE analyses yielded a topology, and the preliminary phylogenetic relationship was hypothesized for the seven genera of Blaptini (Figure 4).

The ML tree revealed that there was a reasonable correlation of membership of each of these major clades to the currently most circumscribed genera. The intergeneric relationships were well supported overall. The individuals of Blaptini were grouped into seven well-supported clades: clade C1 (uBV = 100%), clade C2 (uBV = 96%), clade C3 (uBV = 100%), clade C4 (uBV = 100%), clade C5 (uBV = 100%), clade C6 (uBV = 98%), and clade C7 (uBV = 100%). The clade C1 (*Nalepa*), clade C6 (*Prosodes*), and clade C7 (*Gnaptor*) corresponded to the currently circumscribed genera and were strongly supported (uBV = 100%). The clade C2 contained three species of *Blaps*, as well as the monotypic genus *Ablapsis* (*A. compressipes* Reitter); the clade C3 was also well supported (uBV = 100%) and included the remaining species of the genus *Blaps*.

However, the clade C4 included two samples of *D. bomina* Ren and Li, 2001, and two samples of *Coelocnemodes* (*C. huizensis* Ren and Li, 2001; *C. tibialis* Ren, 2016). Thus, taxonomic status of *D. bomina* Ren and Li, 2001, may need to be re-evaluated. The clade C5 includes the larva sample and two adult samples of *D. ngaria* sp. n. with *D. laevicollis* (uBV = 100%). Based on the above results, the larva sample was confirmed to be conspecific with the adults of *D. ngaria* sp. n. The new species from the western Himalaya is very similar with the type species of *Dila* (*D. laevicollis*) in morphological character (body nearly cylinder-shaped, elongated; antennomeres Ⅲ–Ⅶ long and cylindrical; legs slender, parameres finger-shaped, strongly elongated). Therefore, its taxonomic status was confirmed by both molecular and morphological evidence.

### 3.3. Bionomics

The new species was found in the southwestern Himalayas of the Ngari Prefecture of China. Interestingly, this species is well adapted to semi-arid and arid environments. The adults were found below big stones or a shield in the grass, and were probably feeding on decaying plant roots or leaves (Figure 5).

## 4. Discussion

### 4.1. Taxonomic Remarks

The larvae and adults of Dila ngaria sp. n. were collected in the field; hence, it was rather difficult to judge the larve stage. Skopin noted that the larvae of Dila were similar to the primitive larvae of Blaps from a similar habitat [12]. The larval description above was inferred to be in its final instar stage based on previous descriptions of Blaps and Nalepa. Larval morphology has been found to offer numerous characters for species recognition and has been used to support the close relationships of genera or subtribes [19,20,21,22]. Blaptini is mostly distributed in centraleastern Asia, and comprise about 500 species [23]. However, for the description of larval morphology within Blaptini, available data are mostly from Blaps and several species of Nalepa, Gnaptorina, and Agnaptoria [24,25,26,27,28,29]. The characteristics of larve for most species are unknown due to the lack of specimens. So, we hope to continue collecting more larvae and associating them with their respective adults by rearing and/or sequencing data. We will then solve the relationships of genera or subtribes of the tribe Blaptini through the addition of larval morphological characters.

### 4.2. Phylogeny and Classification of Species

Blaptini is currently divided into five or six subtribes: Blaptina Leach, Gnaptorina Medvedev, Gnaptorinina Medvedev, Prosodina Skopin, Remipedellina Semenow, and Dilina Ren [10,29]. The subtribe Dilina Ren, 2016 (type genus Dila), was erected by Ren in 2016. The position of Dila within Blaptini and the classification of Dilina are debatable because of the presence or absence of apical and ventral teeth on the inner side of the profemora, the structure of the male genitalia, and the close similarity of the larvae of Dila with some Blaps and Nalepa of Blaptina [2,12,30]. Nearly all past contributions focusing on the Blaptini have been based on morphological characters, which did not strongly support the monophyly of the subtribe Dilina [2]. The present work is the first molecular phylogenetic analysis within the Blaptini. The result showed that the classification system of Dila and Dilina are not supported by available phylogenetic data.

The genus Blaps was erected by Fabricius in 1775. All species are flightless and well adapted to semi-arid and arid environments because of several specific behavioral and morphological adaptations [31]. In this study, the phylogenetic relationships of seven species indicated that the genus Blaps is likely paraphyletic. The genus comprises about 250 species, and it would be premature to comment on monophyletic of the genus based on a molecular analysis of only seven species. In order to better understand the genus, subgeneric, and species relationships in Blaptini, we are undertaking additional phylogenetic analyses that will include more species.

In the last revision of Dila by Chigray, four species were transferred from Caenoblaps to Dila with their holotype or lectotype photos provided. Meanwhile, three new species were described from the Hakkary Province of Turkey and West Azerbaijan Province of Iran [2]. Most species of Dila are distributed in medium-high to high elevation mountain habitats. The type species *D. laevicollis* Gebler, 1841, inhabit stony low-elevation mountains and foothills (including the Betpak-Dala desert) and are widespread in Afghanistan, Kyrgyzstan, Kazakhstan, Tajikistan, Turkmenistan, Turkmenistan, and Xinjiang, China [6,7]. Coelocnemodes is an endemic genus of the Tibetan Plateau that is distributed in Pakistan and Yunnan, China. All species of Coelocnemodes have two teeth on the inner side of the profemora and differ from Dila due to a wider body. In contrast, most species of the genus Dila have only one apical tooth on the profemora and slender, elongate bodies. The preliminary phylogenetic results show that the two samples of *D. bomina* Ren and Li, 2001, with C. huizensis Ren and Li, 2001, and C. tibialis Ren, 2016, from Coelocnemodes comprise a single well-supported clade C4 (uBV = 100%); the larva sample and two adult samples of *D. ngaria* sp. n. and *D. laevicollis* are clustered in a single well-supported clade C5 (uBV = 100%). The specimen type and topotypes of *D. bomina* were re-examined: the body is wider; antennomere VII is slightly longer than antennomere VIII; one or two apical tooth exist on the profemora; the male genitalia is C-shaped, the basal piece of the aedeagus is weakly bent, and the apical piece has parameres that are not bent, strongly elongate, and acute. These characters are distinct with the type species *D. laevicollis* and the new species *D. ngaria* sp. n. (body nearly cylinder-shaped; antennomere VII distinctly longer than antennomere VIII; legs slender, profemora with obtuse tooth; male genitalia finger-shaped, apex obtuse, widest at base, evenly narrowed to acute apex), but the morphological characters of D. bomina are very similar with some species of Dila, such as *D. difformis* König, 1906, *D. nitida* Schuster, 1920, *D. baeckmanni* Schuster, 1928, *D. kulzeri* Schuster, 1928, *D. hakkarica* Chigray, 2019, and *D. svetlanae* Chigray, 2019 (body larger or robust, slender; profemur with tooth; basal piece of parameres is wider, evenly narrowed to acute apex, elongated). In addition, *D. bomina* was found from Bowo, Xizang (Eastern Himalayas), with Coelocnemodes being found from a similar habitat. We suspect that the genus Dila is probably non-monophyletic, and the taxonomic status of *D. bomina* is in question upon combining these morphological characters and the results of molecular phylogenetic analysis. However, the molecular data of more Dila species are needed to further explore the above questions.

## 5. Conclusions

In this present study, the larva and adult males and females of a new *Dila* species were described and illustrated based on morphological and molecular evidence. Meanwhile, a preliminary phylogenetic tree of seven related genera of Blaptini suggests that *Dila* is probably a non-monophyletic genus, and the taxonomic status of *D. bomina* Ren and Li, 2001, needs to re-evaluated. The monophyly of the subtribe Dilina is not supported by available phylogenetic data. However, this work provides valuable data for further study on the molecular phylogeny of the tribe Blaptini. We suggest that these incongruences will be more fully resolved by phylogenetic analyses of Blaptini that include more genera, species, and genomic sequences.

## Figures and Tables

**Figure 1 insects-14-00284-f001:**
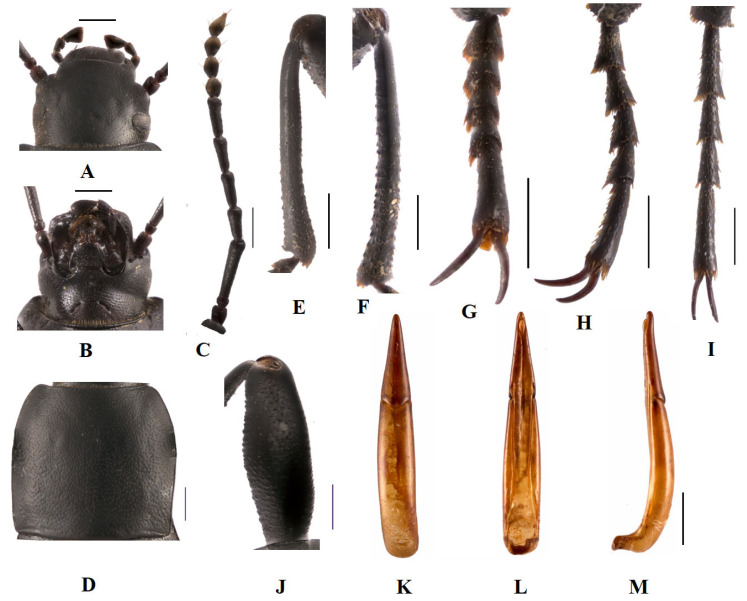
*D. ngaria* sp. n. (**A**). head, dorsal view; (**B**). head, ventral view; (**C**). antenna; (**D**). pronotum; (**E**). profemur; (**F**). protibia; (**G**). mesotibia; (**H**). protarsus (**I**). mesotarsus; (**J**). metatarsus; (**K**–**M**). aedeagus: (**K**). dorsal view; (**L**). ventral view; (**M**). lateral view. Scale bars: 1.0 mm.

**Figure 2 insects-14-00284-f002:**
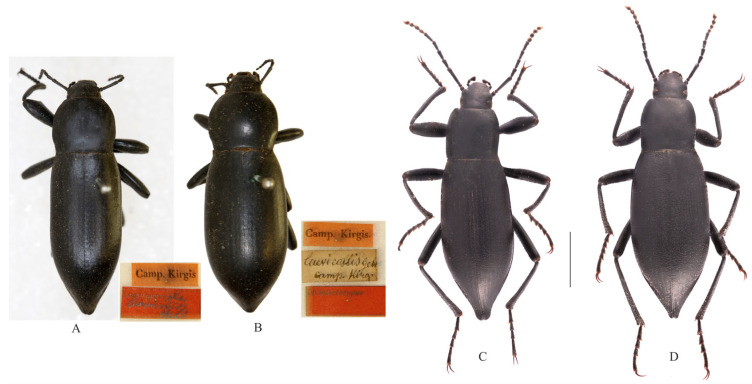
Habitus of the type species and a new *Dila* species. (**A**,**B**). *Dila laevicollis*. (**A**). male, lectotype, (**B**). female, paralectotype; (**C**,**D**). *D. ngaria* sp. n. (**C**). male, holotype, (**D**). female, paratype. Scale bars: 5.0 mm.

**Figure 3 insects-14-00284-f003:**
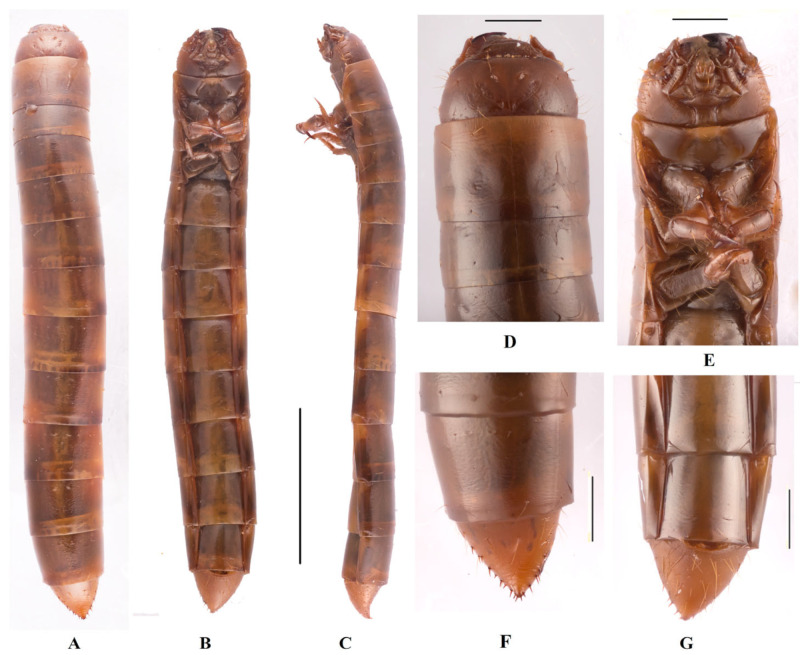
Larva of *Dila ngaria*
**sp. n. A**–**C**. Habitus: (**A**). dorsal view; (**B**). ventral view; (**C**). lateral view. (**D**). head, dorsal view; (**E**). head and prothoracic leg, ventral view; (**F**). pygopods, dorsal view; (**G**). pygopods, ventral view. Scale bars: 5 mm (**A**–**C**), 1 mm (**D**–**G**).

**Figure 4 insects-14-00284-f004:**
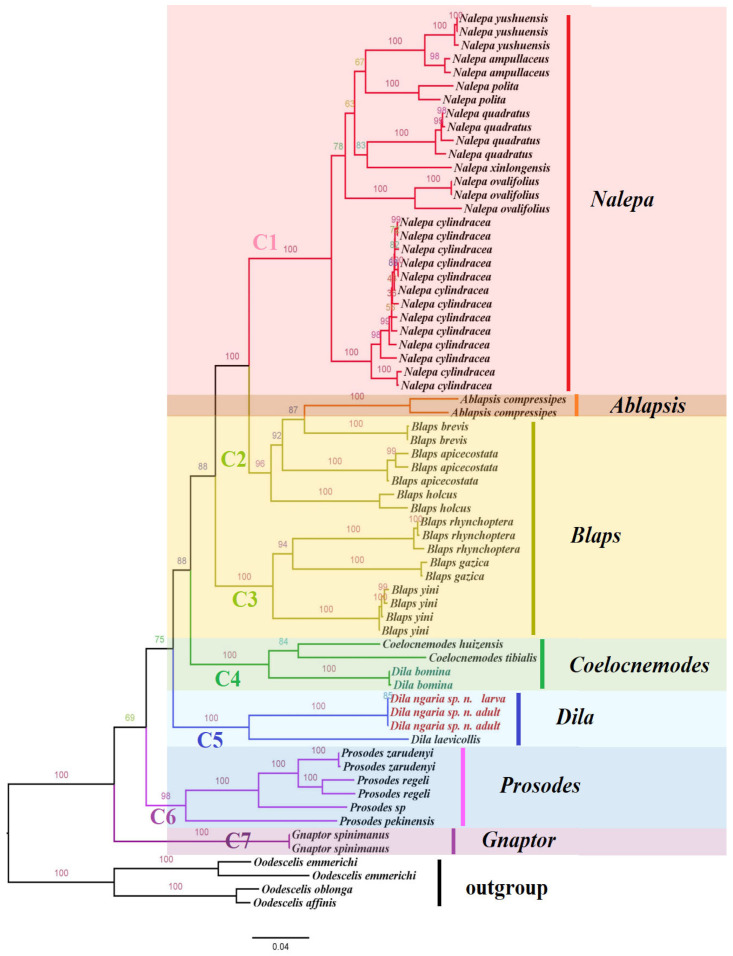
Maximum likelihood phylogenetic tree based on 2289 bp of mitochondrial and nuclear DNA sequences (COI, Cytb, 16S rDNA, and 28S rDNA) within the tribe Blaptini. Support for each node is represented by ultrafast bootstrap values (uBV).

**Figure 5 insects-14-00284-f005:**
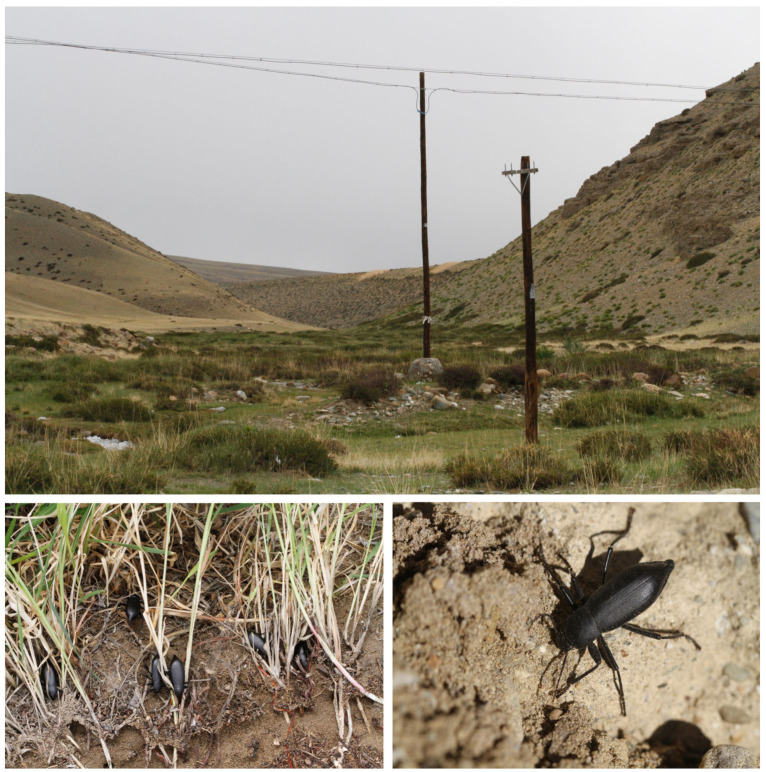
Habitat for *D. ngaria* sp. n. Photo by Xinglong Bai at Zanda of Ngari Prefecture, Xizang, China, on August 2018.

**Table 1 insects-14-00284-t001:** Primer sequences for PCR.

Gene	Primer (Forward/Reverse)	Sequence (Forward and Reverse) 5′ → 3′	PCR Conditions (Annealing)	References
COI	F 2183	CAACATTTATTTTGATTTTTTGG	50 °C	Monteiro & Pierce, 2001 [13]
	R 3014	TCCAATGCACTAATCTGCCATATTA		
Cytb	F revcb2h	TGAGGACAAATATCATTTTGAGGW	50 °C	Simmons et al., 2001 [14]
	R rebcbj	TCAGGTCGAGCTCCAATTCATGT		
16S	F 13398	CGCCTGTTTATCAAAAACAT	50 °C	Simon et al., 1994 [15]
	R 12887	CCGGTCTGAACTCAGATCAT		
28S-D2	F 3665	AGAGAGAGTTCAAGAGTACGTG	58 °C	Belshaw et al., 1997 [16]
	R 4068	TTGGTCCGTGTTTCAAGACGGG		

## Data Availability

The data of the research were deposited at the College of Life Sciences, Hebei University, Baoding, China. The specimens associated with this paper, as well as the sequence data submitted to GenBank, conform to the 2014 Nagoya Protocol on Access to Genetic Resources and the Fair and Equitable Sharing of Benefits Arising from their Utilization (https://www.cbd.int/abs/, accessed on 1 June 2022).

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
