# Peer review of "The Adult and Larva of a New Species of the Genus Dila (Coleoptera, Blaptinae, Blaptini) from Himalayas, with Molecular Phylogenetic Inferences of Related Genera of the Blaptini†"

_insects, 2023, doi:10.3390/insects14030284_

Round 1

Reviewer 1 Report

Dear Editor,

This is a comprehensive paper on the genus Dila with a new species from China. The authors described a new species based on both morphological and molecular evidence, and they also included photographs of the immature stages as well as habitat. All of these sources of information will be useful for future research on this little-studied group of only 22 species in the world.

However, I only have a few minor suggestions, which are highlighted in yellow in the attached pdf.

1.      Every nomenclatural/taxonomic change/new classification should be bold and no comma/full-stop with the author’s name or year as below.

Ablapsis Reitter, 1887 syn. n.

Blaps compressipes (Reitter, 1887) comb. n.

D. ngaria Li & Ren sp. n.

2.      Key to species

A.     The main characters should be separated by a semicolon rather than a full stop.

B.     If you include the figure number for the diagnostic characters, it will be easier to understand them.

C.      It is not necessary to include the helping verbs in the key.

3.      Dila ngaria Li & Ren sp. n.

A.     The majority of the figures represent this species, but you only mentioned one (Figure. 2C-D). It should be updated to include the additional figures (Figure. 1A-M, 2C-D, 3A-G, 5).

4.      Thorax (Figure. 1D).

A.     Instead of a full stop, the main characters should be separated by a semicolon, or the same character of the body part should be separated by a semicolon and begin with a small letter. This is a regular problem in the manuscript and is highlighted in the pdf.

B.     There are some typological mistakes are also highlighted in the manuscript.

5.      Figures/Fig/Figs?

This format is inconsistent throughout the manuscript. Figure legends should follow the same format.

6.      Figure 2.

Leclotype: may be a typological mistake (lectotype)

Hololeclotype: what about this?

7.      Body description

The format should be same. Either add a full stop or colon.

Body (Figs. 3A-C).

Head (Figure. 3D-E).

Thorax (Figure. 3D-E):

Author Response

Q: If you include the figure number for the diagnostic characters, it will be easier to understand them.

Re: We have added the figure number, thanks!

Q: Figure 2. Leclotype: may be a typological mistake (lectotype); Hololeclotype: what about this?

Re: Error has been modified; Hololeclotype: after rechecking, we have changed it to paralectotype.

Q: Instead of a full stop, the main characters should be separated by a semicolon, or the same character of the body part should be separated by a semicolon and begin with a small letter. This is a regular problem in the manuscript and is highlighted in the pdf.

Re: We have revised the manuscript according to reviewer comments and advice. Additional corrections have also been made on some grammar and spelling errors.

Reviewer 2 Report

The article reflects research conducted mainly in genetic terms, with a serious omission of the morphological analysis necessary in the case of taxonomic work, in which the typical material of the species subject to synonymization should be analysed.

I consider it necessary to analyze the type specimens of synonymized species, to present a list (table or photos) of morphological features and appropriate photographs or drawings, showing the basis for the synonymization of Blaps and Ablapsis, and thus, new combinations.

It should be remembered that in taxonomic works, regardless of the methods used, it is necessary to apply the recommendations of the International Code of Zoological Nomenclature and, most importantly, to analyze the type specimens of the studied species and genera, primarily if their synonymization is carried out. What's more, in the material and methods part, you should present the data written on the labels of the analyzed specimens.

Author Response

Q: on some grammar and spelling errors

Re: We have revised the manuscript according to reviewer comments and advice. Additional corrections have also been made on some grammar and spelling errors.

Q: Please give a ratio - slender, eolongate, robust and weekly elongate is a very relative description, especially in the key to the species (including one new species)

Re: We have added the figure number for the diagnostic characters, it will be easier to understand them.

Q: For D. ngaria Li & Ren, sp. n.  Did you checked any other species, not from China, which could be distributed in other countries around China?

Re: We have read the following paper through carefully before confirming new species.

Chigray, I.A.; Nabozhenko, M.; Abdurakhmanov G.; Keskin B. A systematic review of the genus Dila Fischer von Waldheim, 1844 (= Caenoblaps, syn.n.) (Coleoptera: Tenebrionidae) from the Caucasus, Turkey and boundary territories of Iran. Insect Systematics & Evolution 2019, 99, 914-923.

Q: Figure 2: Data from the labels shoud be in Material and methods part;hololeclotype:Holotype or Lectotype?

Re: We have added label information in material and methods part, thanks!

   Hololeclotype: after rechecking, we have changed it to paralectotype.

Q: The article reflects research conducted mainly in genetic terms, with a serious omission of the morphological analysis necessary in the case of taxonomic work, in which the typical material of the species subject to synonymization should be analysed.

I consider it necessary to analyze the type specimens of synonymized species, to present a list (table or photos) of morphological features and appropriate photographs or drawings, showing the basis for the synonymization of Blaps and Ablapsis, and thus, new combinations.

It should be remembered that in taxonomic works, regardless of the methods used, it is necessary to apply the recommendations of the International Code of Zoological Nomenclature and, most importantly, to analyze the type specimens of the studied species and genera, primarily if their synonymization is carried out. What's more, in the material and methods part, you should present the data written on the labels of the analyzed specimens.

Re: the synonymy is proposed: Blaps Fabricius, 1775 = Ablapsis 317 Reitter, 1887 syn. n. We are also considering a little imprudent based on the available data. So, we deleted this section of the discussion and it would be premature to comment monophyletic of the genus based on a molecular analysis of only few species. In order to better understand generic relationships of Blaptini, we are undertaking additional phylogenetic analyses in another paper that will include more species of Blaps (more than 100 species).

Round 2

Reviewer 2 Report

The paper has been corrected according to the suggestion.

Author Response

The paper has been corrected according to the suggestion.